# A Lightweight Browser-Based Tool for Collaborative and Blinded Image Analysis

**DOI:** 10.3390/jimaging10020033

**Published:** 2024-01-27

**Authors:** Philipp Schippers, Gundula Rösch, Rebecca Sohn, Matthias Holzapfel, Marius Junker, Anna E. Rapp, Zsuzsa Jenei-Lanzl, Philipp Drees, Frank Zaucke, Andrea Meurer

**Affiliations:** 1Department of Orthopedics and Traumatology, University Medical Center of the Johannes Gutenberg, University Mainz, 55131 Mainz, Germany; philipp.drees@unimedizin-mainz.de; 2Department of Orthopedics (Friedrichsheim), University Hospital Frankfurt, Goethe University, 60528 Frankfurt am Main, Germany; marius.junker@web.de; 3Department of Orthopedics (Friedrichsheim), Dr. Rolf M. Schwiete Research Unit for Osteoarthritis, University Hospital Frankfurt, Goethe University, 60528 Frankfurt am Main, Germanya.meurer@medicalpark.de (A.M.); 4Department of Orthopedics, Tabea Hospital Hamburg, 22587 Hamburg, Germany; 5Medical Park St. Hubertus Klinik, 83707 Bad Wiessee, Germany

**Keywords:** collaborative blinded image analysis, randomization, anonymization, multiple observers

## Abstract

Collaborative manual image analysis by multiple experts in different locations is an essential workflow in biomedical science. However, sharing the images and writing down results by hand or merging results from separate spreadsheets can be error-prone. Moreover, blinding and anonymization are essential to address subjectivity and bias. Here, we propose a new workflow for collaborative image analysis using a lightweight online tool named Tyche. The new workflow allows experts to access images via temporarily valid URLs and analyze them blind in a random order inside a web browser with the means to store the results in the same window. The results are then immediately computed and visible to the project master. The new workflow could be used for multi-center studies, inter- and intraobserver studies, and score validations.

## 1. Introduction

### 1.1. Challenges in a Standard Workflow for Image Analysis

Reliable manual image analysis is essential to scientific work but has several challenges. Among these are subjectivity and bias. They occur, for instance, when the observer can view metadata, e.g., a patient’s name and date of birth and the date of acquisition in radiologic images. However, even simple file names bear the risk of bias during image analysis due to known group allocations that might be disclosed even by the specimen ID. Thus, pseudonymization does not fully address subjectivity and bias. In daily practice, images are often distributed for analysis by sharing large file folders via Clouds, E-Mail, or USB devices accompanied by an additional spreadsheet for evaluation and later statistical analysis. In clinical settings, observers are often provided with a list of patients, and images are analyzed using the hospital’s local picture archiving and communication system (PACS), where patient data are visible, possibly influencing the analysis [1,2]. Hence, to minimize subjectivity and bias, any person included in the study and data analysis should be blinded [3] and the data analyzed in a randomized order. That is why journals such as *Nature* request information about blinding and randomization in studies, including if and how randomization was achieved and whether data analysis was performed blinded [4].

Another challenge is the collaborative analysis by multiple observers. This is especially important since one observer’s assessment might produce subjective results, while the sum of multiple opinions tends to produce more objective results. Experts from outside their own work group/institution should ideally be included; however, transferring the data, explaining the approach, and gathering the results can be laborious and complicated. Finally, regardless of their location and the use of hard- or software, all observers within one study should use the same standardized analysis tools (i.e., software version). It should be remembered that all steps involving multiple observers can be laborious and time-consuming; this includes preparatory steps such as (pseudo)-anonymization, as well as the gathering and merging of data after analysis via copy-and-paste or by hand, all of which are error-prone [5,6].

### 1.2. Available Tools for Image Analysis

Various tools with differing advantages and limitations are available to analyze images [7]. Amongst these, tools like ImageJ/Fiji (version number) (open source) [8,9] or QuPath (open source) [10] comprise potent tools for creating complex image analysis pipelines [11]. Medical images, including X-rays, CTs, or MRIs, are usually visualized with essential measurement tools on PACS-viewers. In contrast, more complex measurements (e.g., joint replacement planning) are performed by exporting the images to specialty-specific tools like mediCAD (mediCAD Hectec GmbH, Altdorf, Germany) in orthopedics.

Another essential entity are image annotation tools. Such tools can be used to provide training data for machine learning algorithms. Tools like Redbrick.ai (Zantula, Inc., Claymont, DE, USA) or Zillin.io (Adaptive Vision, Gliwice, Poland) allow multiple observers to annotate anonymized images [12,13]. Apart from various annotation utensils like brushes, pens, etc., Zillin.io, for instance, allows the project master to create single-choice questions. Hence, Zillin.io provides a blinded image assessment. However, at the time of writing, it does not include tools to measure a length or an angle. Redbrick.ai and Zillin.io promote a collaborative approach by displaying images inside a web browser.

To score images blinded, a tool named Blinder (Steven Cothren, Solibyte Solutions LLC, Durham, NC, USA) is freely available. Blinder is desktop software that displays images in a random order while hiding metadata. A definable single-choice question with answers displayed as buttons is shown next to the image. Thus, Blinder allows the blinded visual scoring of images [14]. However, Blinder does not display medical images in DICOM format or provide measurement tools. Furthermore, since Blinder is desktop-bound, including multiple observers from different locations is challenging. 

### 1.3. How Is Image Analysis Currently Performed?

A systematic review was conducted to assess the status quo in scientific image analysis and, especially, if and how anonymization and randomization are performed. Therefore, the PubMed database was screened for articles utilizing a typical histologic score. The Osteoarthritis Research Society International (OARSI) score was chosen [15]. Using the keyword “oarsi score”, free full-text articles from 2022 with an impact factor (IF) equal to or higher than five were selected on 17 December, 2022. Nineteen articles (Appendix A) with a mean IF of 6.1 ± 0.9 were retrieved [16,17,18,19,20,21,22,23,24,25,26,27,28,29,30,31,32,33]. One study was not included since it did not apply the OARSI histopathological score but a performance test with the same name. Of the 18 authors, 12 (67%) stated that they had analyzed images using the OARSI score with multiple blinded observers. The remaining authors did not adhere to this protocol or did not provide information on how many observers analyzed the images and whether they were blinded. None of the 18 authors offered detailed information on how the images were anonymized and how the blinded analysis by multiple observers was conducted specifically. None of the authors mentioned using collaborative software or any form of AI for this process. In summary, image analysis by numerous blinded observers appears to be a commonly utilized approach. However, the lack of detailed information on how the research is conducted leads to the conclusion that the process is still not standardized or homogeneous. This might lead to results with different levels of objectivity, i.e., quality, and might make results from other authors less comparable. Therefore, we conclude there is a need for a tool that standardizes and facilitates blinded image analysis by multiple observers.

### 1.4. A Standard and a New Workflow for Image Analysis

Without using specific software, a standard workflow for manual image analysis by multiple observers usually follows a particular protocol (Figure 1A): (I) Images are selected and pseudonymized. A spreadsheet with questions and answers or tasks is created. (II) The images and the spreadsheet are shared with observers; this can be performed via E-Mail, USB devices, or Cloud services. (III) The observers use desktop-bound software to assess and analyze the images; results are stored in the provided spreadsheet. (IV) The observers return their spreadsheets to the project master. (V) The project master merges all results and prepares the data for statistical analysis. 

Since the standard workflow might be error-prone, time-consuming, and contains the risk of subjectivity and bias, an alternative workflow using a novel software, Tyche, is proposed. Tyche is a browser-based tool that displays anonymized images in a random order with standard tools to assess and analyze the images. Observers can gain access with temporarily valid and project-specific URLs. Customizable input forms and buttons are used on the same window to store results directly. Hence, the results are immediately merged, computed, and visible to the project master. Thus, the new workflow contains the following steps (Figure 1B): (I) Images are selected and anonymized; they are uploaded to Tyche, and tasks and questions with answers are created online. (II) Observers are provided with a project-specific URL. (III) Observers assess and analyze the images inside their web browser in a random order without access to meta-data. (IV) Results are immediately merged, computed, and visible to the project master.

## 2. Materials and Methods

### 2.1. Comparison of a Standard and New Workflow with an Exemplary Image Analysis Scenario

A typical image analysis scenario was replicated to compare the standard with the proposed workflow using Tyche. As described in the systematic review (see introduction), histopathologic images were scored using the OARSI score. Four experienced researchers were asked to analyze twenty-four images, once using the standard workflow and once using Tyche. Images were selected from our previous research [34,35], including samples from 2, 4, 8, and 12 weeks after intervention. For the standard workflow, images were pseudonymized by naming them in ascending order from 1 to 24. An additional spreadsheet stored information regarding which image belonged to which group. Using Tyche, four data groups (“2”, “4”, “8”, and “12”) were created, and images were uploaded directly via drag-and-drop into these groups. No pseudonymization was needed. For the offline approach, observers were provided with the images via E-mail, including a spreadsheet to note results. The same observers received another E-mail with the project-specific URL to analyze the images using Tyche. Observers received both E-Mails at the same time. Without giving further notice to the observers, the time needed to provide the project master with all results from both workflows was recorded and compared.

### 2.2. Creating a New Project for an Exemplary Image Analysis with Tyche

To start a new image analysis with Tyche, the following steps are used to create a project: (I) defining the project name, (II) creating questions/tasks (Figure 2A), and (III) uploading images (Figure 2B). Questions or tasks can be individually created for each project and must be answered by each observer for every image with a single choice or numeric response. Image files can be uploaded via drag-and-drop and, if needed, assigned to different data groups, e.g., Placebo vs. Verum or two weeks vs. four weeks. If desired, observers can choose an observer group at the beginning of their analysis (e.g., PhD student or Postdoc, etc.) that reflects their experience level. The project master can define these observer groups at the creation of the project. Once a project is created, it appears in the user’s project list. From here, access to the data can be shared via project-specific, temporary valid URLs with observers; moreover, the results can be displayed (Figure 2C).

### 2.3. Analyzing Images Using Tyche

As illustrated in Figure 3, Tyche can display DICOM images, such as X-rays. Standard tools like zoom, pan, and contrast can be used for analysis, including count, length, angle, freehand area, and probe. In addition, Tyche recognizes length-per-pixel ratios stored as meta-data in DICOM images and translates measurements from pixel (px) to length (mm) and area (mm^2^) accordingly. Alternatively, Tyche provides a calibration tool that allows one to enter a scaling factor directly or calculate the scale by measuring the current size (in px or mm) of an object with a known length, e.g., a ball marker (usually 25 or 30 mm) in orthopedic imaging (Figure 3B). Histologic images stored as JPG or PNG can also be analyzed (Figure 3C,D) using the abovementioned tools.

### 2.4. Displaying Results with Tyche

Once an observer finishes the analysis, the results are immediately visible to the project master (Figure 4). The results can be combined or displayed for the individual observer, including standard deviations. In addition, the project master can see a list of all the images and how the observers analyzed each image. Moreover, results can be divided according to observer groups (for example, PhD student vs. Postdoc) or data groups (for example, Placebo vs. Verum) defined by the project master when creating the project. Finally, results can be exported in CSV format by clicking the “export” button.

### 2.5. Technical Aspects of Tyche

Tyche employs the latest versions of several open-source libraries and software: Joomla! (Open Source Matters, New York, NY, USA, GNU General Public License), a content management system is used for user management. dropzoneJS (Matias Meno, MIT License) facilitates uploading the data. Tyche can manage and display images (*.jpg, *.png), videos (*.mp4), and DICOM files (*.dcm), as well as zip-compressed stacks of the latter. Using the cornerstone library (Chris Hafey, MIT License), images are rendered into an HTML canvas. Thus, they cannot be accessed directly, nor can their location on the server be retrieved. Instead, every project is stored with all relevant information inside a MySQL table (Oracle Corporation, Austin, TX, USA). From there, images are selected for display using the MySQL “select random” function. All connections are encrypted via the Secure Sockets Layer (SSL). The server hosting Tyche is located in Europe. The name Tyche refers to the ancient Greek goddess of fortune and destiny [36].

## 3. Results

In this study, we outlined the need for and creation of a new workflow for collaborative and blinded image analysis. We replicated a standard image analysis scenario to compare a standard workflow with the proposed workflow in order to identify the differences and possible benefits. The new workflow utilizes a browser-based tool, Tyche, that displays images in a random order inside a web browser with means to store results online in the same window. Tyche thus promotes collaborative and blinded image analysis.

We asked four experienced researchers to analyze twenty-four tissue sections from murine knee joints in which osteoarthritis was surgically induced by the destabilization of the medial meniscus [37]. The medial tibial cartilage was evaluated according to the recommendations of the Osteoarthritis Research Society International (OARSI) histopathology initiative [15] using a score with grades from 0 to 6 (Figure 5A). The images were assessed using a standard workflow in which observers were provided with the images and a spreadsheet via E-Mail. In addition, the same images were evaluated by the same observers using the new workflow by employing Tyche. Thus, observers received another E-Mail with a project-specific URL to analyze the images inside their web browser. Observers received both E-Mails at the same time. Without giving further notice to the observers, the time needed to provide the project master with all results was compared. Results were available significantly faster using the new workflow (84.3 ± 35.5 h vs. 15.3 ± 12.7 h, Mann–Whitney test, *p*-value = 0.03, h = hours). The results did not differ significantly between the workflows (Figure 5B) and correlated well (Figure 5C).

## 4. Discussion

### 4.1. Differences between the Workflows

While the final results did not vary between the two workflows, important differences were noted during the analysis: Using Tyche, the images were analyzed, anonymized, and organized in a random order, while using the standard workflow, observers were able to see the filename, view multiple images at the same time, and go back and forth between the images. In contrast, with Tyche, there was no need to anonymize the images. With the standard workflow, at best, the analysis can be pseudo-anonymized, and filenames can be replaced with an ID, as was done in this study. Using Tyche, observers were completely blinded and could not move back and forth between images or open multiple images simultaneously. Even though complete anonymization and blinding did not provide significantly different and, thus, potentially better results in the performed analysis, it is a generally preferred approach [4]. Another notable difference between the workflows was usability. With Tyche, observers did not have to store results on handwritten notes or in spreadsheets. Instead, the results were immediately stored online and available after the analysis. These aspects simplify the analysis, might reduce the risk of errors, and speed up the entire process. The fact that the observers can perform the analysis wherever and whenever desired could increase the willingness of experts to participate in a study and shorten the time until the analysis is finished. While it is generally challenging to quantify usability, a time comparison in this study showed that results were obtained faster using the new workflow. Tyche addresses potential subjectivity and bias in the standard workflow by showing images in an anonymized manner and a random order. 

### 4.2. Advantages and Disadvantages of the New Workflow

Tyche offers essential tools for analysis like count, length, surface, angle, and color measurements. However, when more complex measurements need to be performed, tools with more sophisticated measurement capabilities might still be required. Tyche was designed to be lightweight and user-friendly. Images do not have to be opened one by one or be searched for in a database like PACS. In contrast, clicking the “Next” button stores results and leads to the following image. However, it has to be noted that Tyche does not facilitate the analysis of the individual image itself. Tyche does not include automated tools that support the observer. Hence, when extensive datasets must be analyzed, tools that offer semi- or fully automatic analysis, for example, Machine Learning, might be a better solution. Still, in this setting, Tyche could be useful as results from an algorithm could be validated using Tyche with a sample-sized subset of the images. In summary, Tyche is useful when using multiple blinded experts to analyze images is more important than analyzing a large number of images.

### 4.3. Settings and Study Designs Where the New Workflow Could Be Used

Tyche is appropriate for standard measurements like length, angle, or area. Machado et al., Micicoi et al., and Schippers et al. have already used it on small series (<200 images) for orthopedic measurements [38,39,40]. In addition, with the single-choice question format, Tyche can be efficiently used to score images, like in a study by Schippers et al., which applied the Kallgren and Lawrence score to more than one hundred pelvic X-rays [41]. Thus, it can be used to determine an existing score on a set of images or validate a new score, as Schippers et al. did to apply a new scoring system for toe alignment in foot surgery [42]. Since Tyche facilitates assessments by multiple observers, multi-center, as well as inter- and intrarater studies, can easily be performed. Especially in multi-center studies, blinding can be an essential and costly challenge; hence, cost-effective and efficient tools are needed [43].

### 4.4. Limitations

This study has some limitations that need to be considered. Comparing the offline method with Tyche, it is essential to note that the performed experiment does not prove the superiority of either approach. The results correlated well, and standard deviations did not differ, indicating that randomization and anonymization did not produce different or potentially more accurate results. Consequently, one might challenge the necessity of a tool like Tyche, and its usability was only quantified by a time comparison in one experiment. However, quantifying a reduction, for instance, in transmission errors, is hard to perform.

Further independent experiments with time comparisons might be necessary as well. Another limitation is the relatively small number of images being used and the fact that only histopathological images were analyzed. Tyche’s performance must be evaluated on larger imaging sets and various imaging types.

## 5. Conclusions

In summary, we present a new workflow for collaborative and blinded image analysis and validate its applicability in the setting of histopathological image analysis. This new workflow is implemented via a novel, free, lightweight, browser-based tool, Tyche, in an attempt to address current challenges in manual image analysis. From a technological standpoint, Tyche functions as an online image viewer that displays anonymized images from a database in a random order. Multiple observers can analyze the images simultaneously with essential imaging tools and store results directly online. Tyche is innovative less in terms of technological aspects and more in terms of usability. At its core, Tyche functions like an online image questionnaire that facilitates blinded image assessment by multiple observers in different locations. Thus, it might help reduce errors, subjectivity, and bias and might make results more reliable and comparable. The true value of the newly developed workflow and Tyche might be best assessed through its adoption and utilization by the scientific community in imaging projects. Empirical studies conclusively demonstrating its superiority over the standard workflow, especially in terms of the reduction of errors, subjectivity, and bias, may be challenging to perform.

## Figures and Tables

**Figure 1 jimaging-10-00033-f001:**
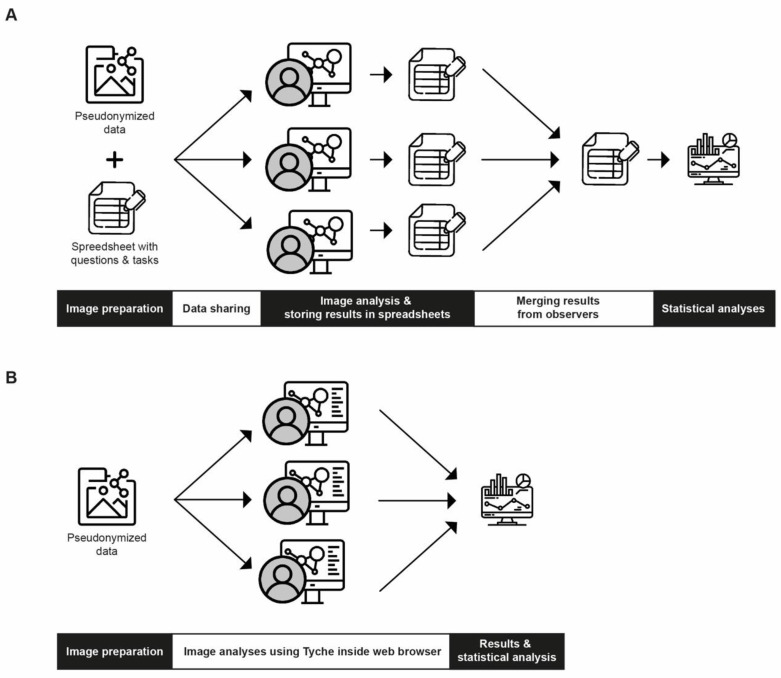
Comparison between an exemplary standard workflow and the proposed workflow using Tyche. (**A**) Exemplary workflow for standard image analysis by multiple observers: Images and a spreadsheet with questions and tasks are prepared and shared with observers via E-Mail, Cloud, or USB devices. Observers perform the analysis using desktop-bound software and store results inside the provided spreadsheet. The project master collects the spreadsheets and merges results for statistical analysis. (**B**) Workflow for collaborative image analysis using Tyche: Images are anonymized and uploaded to Tyche. Tasks and questions are created. Observers are provided with a project-specific URL and are required to analyze the images inside their web browser and store results in the same window. After analysis, the results are immediately merged, computed, and visible to the project master. They can be exported for further statistical analysis. (Credits for icons: see Appendix A).

**Figure 2 jimaging-10-00033-f002:**
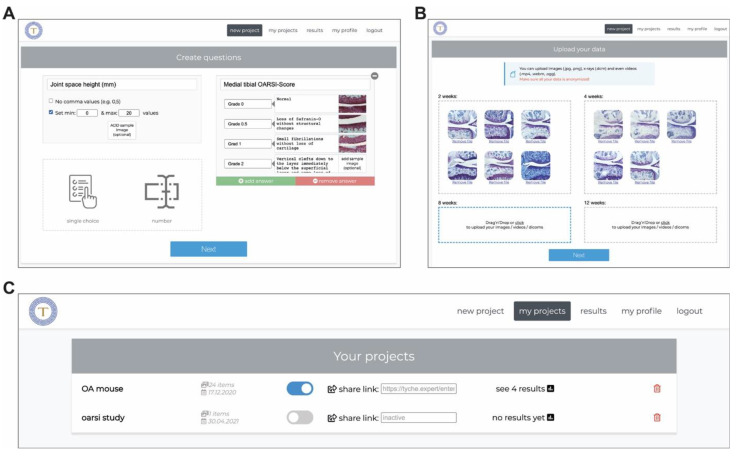
Creation of a new project for image analysis in Tyche (**A**). Tasks and single-choice questions with answers can be tailored for the analysis. (**B**) Images can be uploaded via drag-and-drop, optionally into different groups as shown here (“2 weeks”, “4 weeks”, “8 weeks”, and “12 weeks”). (**C**) List of projects on the homepage of a Tyche user. Each project has a toggle switch to enable or disable the share link, i.e., the URL to access the data. In addition, the number of results is shown. (Source: Screenshot from Tyche interface, taken by Tyche’s first author and creator. Histologic images used with permission from Rösch et al., 2021 [34]).

**Figure 3 jimaging-10-00033-f003:**
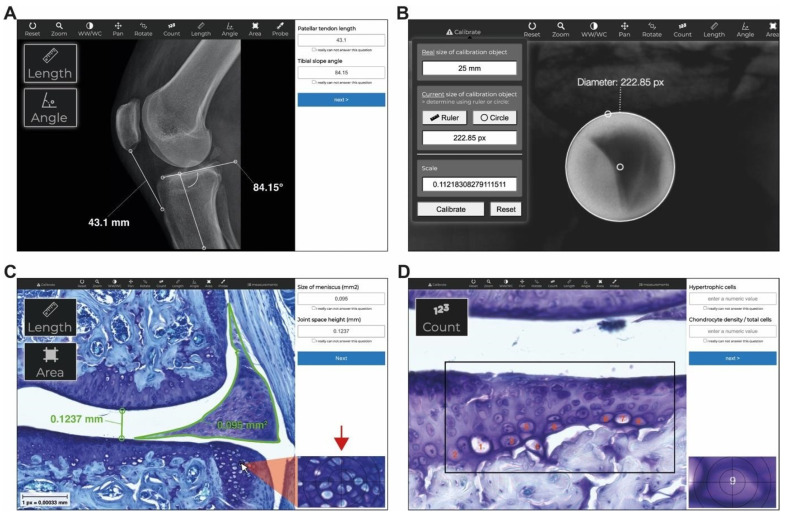
Analysis of images using Tyche. Tyche can display different formats of images like X-rays in JPG or DICOM format (**A**) or histologic images (**C**,**D**). DICOM images can be calibrated to perform length measurements (**B**). On the right side of the screen, results can be stored immediately inside tailored customizable input forms. The red arrow in (**C**) shows a magnification of the underlying image at the position of the cursor. The black frame in (**D**) shows the area where observers counted cells. (Source: A: Knee X-ray from our database obtained with informed consent of the patient. (**C**,**D**) Histologic images used with permission from Rösch et al., 2021 [34]).

**Figure 4 jimaging-10-00033-f004:**
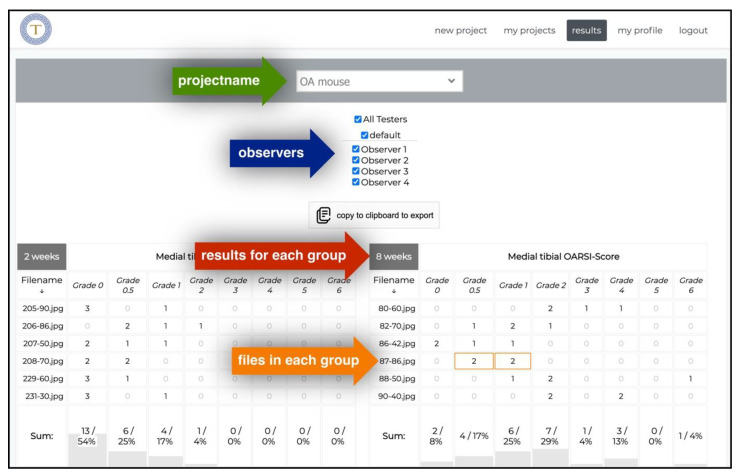
Displaying results online (only visible to project master). After analysis, results are immediately visible online. Results from a project (green arrow) can be shown combined or filtered for individual observers (blue arrow). Results are directly merged and computed for different data groups defined during the project’s creation (red arrow). Results can be traced back to individual images (orange arrow). The orange box shows exemplary results for image “87–86.jpg”: two observers answered with “Grade 0.5”, and two observers answered with “Grade 1”.

**Figure 5 jimaging-10-00033-f005:**
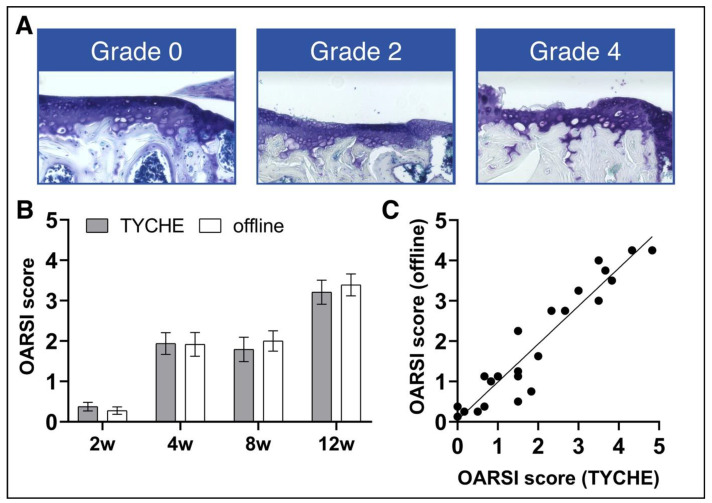
Analyzing the OARSI score using a standard (offline) and new workflow (Tyche). Representative pictures of the medial tibial articular cartilage contact area using histological staining (**A**) and the quantitative analysis of the OARSI score using Tyche and the standard workflow (**B**), as well as the Spearman’s correlation (r = 0.951) between the two approaches (**C**). (Source: Histologic images used with permission from Rösch et al., 2021 [34]).

## Data Availability

The data presented in this study are available on request from the corresponding author.

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
