# Peer review of "A Lightweight Browser-Based Tool for Collaborative and Blinded Image Analysis"

_2313-433X, 2024, doi:10.3390/jimaging10020033_

Round 1
Reviewer 1 Report
Comments and Suggestions for Authors
In this paper, the author present a new workflow for the collaborative and blinded image analysis. However, there are several issues for the author to improve.
1. The author should figure out what is the significance of this study? How would this research change the existing workflow if apply to the clinic?
2. What are the efficiency of their workflow to existing standards? Are there any literatures to support their workflow?
3. Since most of imaging devices have their own software, how would their workflow address this issues?
Reviewer 2 Report
Comments and Suggestions for Authors
Researchers have developed a new workflow for collaborative and blinded image analysis using a lightweight online tool called Tyche. This workflow addresses the challenges of sharing images, writing down results, and ensuring blinding and anonymization for manual image analysis by multiple experts in different locations. The new workflow allows experts to access images via temporarily valid URLs, analyze them in random order and blinded inside a web browser, and store results directly on the same window. Results are then immediately computed and visible to the project manager. This workflow could be used for multi-center studies, inter- and intraobserver studies, and score validations.
Dear authors, it was pleasure to read your interesting manuscript, however I am affraid it might have a fundamental flow which can be possibly explained and there are also some opportunities to make it more readable for your readers.
Here are 5 critical things I believe can be improved in your manuscript:
1. Improve the title. The current title is too generic and doesn't accurately reflect the content of the paper. A better title would be something like "A Lightweight Browser-Based Tool for Collaborative and Blinded Image Analysis". I leave it up to you and I dont insist on this. Also in comparison to your other publications regarding Tyche, you can consider to reffer them in the paper more in deeper context.
2. Provide more fundamental details about Tyche. Your manuscript should provide more details about the features and capabilities of Tyche. Despite you mention that the online tool Tyche is free I do not understand why you dont provide the link where it is available at all. If it is this: https://www.tyche.expert/ you shall provide at least a temporary testing account access for reviewers. Otherwise your proclamations are not properly verifieble. Also you should specify the types of images that Tyche can analyze in more detail, the measurement tools that it provides, and the features that support collaborative analysis. Unless this is clarified I can not recommend this paper for publication.
3. Strengthen the comparison of the standard workflow and Tyche. The paper should provide a more rigorous comparison of the standard workflow and Tyche. This could include a larger sample size, a more diverse range of images, and a more comprehensive assessment of the potential benefits and drawbacks of each approach.
4. Address the limitations of the study. The paper should acknowledge the limitations of the study and suggest potential areas for further research. For example, you could mention that the study was limited to a small sample of images and that further research is needed to evaluate the performance of Tyche on a wider range of image types.
5. Improve the clarity of the writing. The paper is well written overall, but there are a few places where the writing could be made clearer. For example, you could provide more specific examples to illustrate your points.
Comments on the Quality of English LanguageIn general, it is fine. Here are some specific suggestions for improving the writing:
- Replace the phrase "essentially a tool" with a more concrete description of Tyche's functionality.
- Use more specific language to describe the benefits of Tyche, such as "reduces errors, subjectivity, and bias" and "makes results more reliable and comparable".
- Replace the phrase "However, usability and usefulness have to be demonstrated in more extensive imaging studies" with a more specific statement about the need for further research
Reviewer 3 Report
Comments and Suggestions for Authors
In this study, the authors proposed a novel workflow for image analysis in an unbiased, collaborative, and fast manner. The workflow is deployed on web and allows users to view, annotate, measure, and grade medical images. While it is widely acknowledged that the advancement in imaging techniques has revolutionized the research in human diseases, there is a significant lag in complementary analytical approaches. Analyses such as this paper represent an important effort to bridge this gap promptly, therefore I recommend the publication of this study. With that said, I do have a few concerns before it can be accepted:
1. The authors mentioned that the URL for accessing the images is temporarily valid. Please specify the exact duration of validity. Is it fixed or changed? What if the URL expired but the observer hasn't finished editing? What if the observer would like to revisit a previous edit for correction but link is expired? The authors should include discussion on how to solve uncertainty.
2. The authors mentioned that the new workflow significantly reduced the time compared to the standard approach. I wonder what factor caused such a dramatic difference?
3. In digital pathology, many more steps are followed by image annotation. Such as image segmentation, registration, etc. The authors indeed mentioned that the results can be exported into CSV format. However, how about the images themselves? Can they be exported as well? If yes, in what resolution?
Minor:
I suggest the following ref, which includes a typical workflow of image analysis in digital pathology:
Mi H, Gong C, Sulam J, et al. Digital pathology analysis quantifies spatial heterogeneity of CD3, CD4, CD8, CD20, and FoxP3 immune markers in triple-negative breast cancer[J]. Frontiers in physiology, 2020, 11: 583333.
Reviewer 4 Report
Comments and Suggestions for Authors
The paper presents Tyche, a new workflow for collaborative and blinded image analysis. The authors claim that Tyche can reduce the time and errors involved in the traditional method. However, the functionality of Tyche seems very limited. Many hospitals have been using electronic medical record (EMR) systems for a long time, which may offer similar or better features than Tyche. I suggest that the authors improve Tyche by supporting more image formats, applying machine learning methods to analyze the images and provide preliminary diagnoses.
Round 2
Reviewer 1 Report
Comments and Suggestions for Authors
No further comments
Reviewer 2 Report
Comments and Suggestions for Authors
authors have revised the manuscript sufficiently, thank you
Comments on the Quality of English Languageis fine
Reviewer 3 Report
Comments and Suggestions for Authors
The authors have addressed all my concerns, I do not have further comments.
Reviewer 4 Report
Comments and Suggestions for Authors
The authors carefully addressed the points raised by the previous reviewers and, from my side, I don't have any additional observation. The manuscript, in my opinion, is ready for publication.